A375 melanoma-derived lactate controls A375 melanoma phenotypes by inducing macrophage M2 polarization via TCA cycle and TGF-β signaling

http://orcid.org/0000-0002-0863-7730 Wang Qifei
Shi Yurui
Qin Zelian
Xu Mengli
Wang Jingyi
Lu Yuhao
Zhao Zhenmin zzmbysy@sina.com
Bi Hongsen bihongsen@bjmu.deu.cn
Department of Plastic Surgery, Peking University Third Hospital , Haidian District, Beijing , China
Sotelo-Mundo Rogerio
Electronic publication date: 2025 Feb 21
Publication date: 2025
Volume: 13
Electronic Location ID: e18887
Received 2024 Jan 31; Accepted 2024 Dec 31
Copyright: © 2025 Wang et al.
Copyright year: 2025
Copyright holder: Wang et al.
License: This is an open access article distributed under the terms of the Creative Commons Attribution License, which permits unrestricted use, distribution, reproduction and adaptation in any medium and for any purpose provided that it is properly attributed. For attribution, the original author(s), title, publication source (PeerJ) and either DOI or URL of the article must be cited.
License URL: https://creativecommons.org/licenses/by/4.0/

Keywords: Lactate, Macrophage, Melanoma, M2 polarization, TCA cycle, TGF-β

Funding: National Natural Science Foundation of China 82172216 and 82202453 This work was supported by National Natural Science Foundation of China [grant numbers 82172216 and 82202453]. The funders had no role in study design, data collection and analysis, decision to publish, or preparation of the manuscript.

==============================
Introduction

Macrophage phenotypes have been linked to progression and prognosis of cutaneous melanoma. However, the association between Warburg effect in A375 melanoma and macrophages polarization, as well as the underlying mechanisms, remains less well documented.

Objective

The present study aimed to investigate the effect of lactate derived from A375 melanoma on macrophage polarization, melanoma phenotype responses and the underlying mechanisms.

Methods

Flow cytometry was performed to evaluate the expression of M1 and M2 markers, cell cycle and apoptosis. Levels of transforming growth factor β (TGF-β) and tumor necrosis factor α (TNF-α) were determined with enzyme-linked immunosorbent assay (ELISA) kit. Proliferation and invasion were assessed by CCK8 and transwell assays, respectively. The extracellular acidification rate (ECAR) and oxygen consumption rate (OCR) were analyzed using an XF96 extracellular flux analyzer. Protein expressions were determined by Western blotting.

Results

Our results revealed that melanoma A375 conditioned medium (A375-CM) induced peripheral blood mononuclear cells (PBMCs) to polarize toward anti-inflammatory M2 macrophages. M2 markers CD206 and ARG1 expression increased, as did TGF-β secretion. Conversely, M1 marker CD68 expression decreased. Furthermore, hypoxia promoted macrophage M2 polarization induced by A375-CM. Elevated lactate level in PIG1-conditioned medium (PIG1-CM) induced M2 polarization, whereas the lactate transport inhibitor AZD3965 suppressed this effect in PBMCs cultured with A375-CM. Additionally, lactate derived from melanoma regulated M1/M2 polarization by the tricarboxylic acid (TCA) cycle instead of glycolysis. Significantly, polarized macrophages altered melanoma phenotypes including proliferation, clone formation, cell cycle, apoptosis, migration and invasion via TCA cycle and TGF-β.

Conclusion

Our data collectively demonstrate that lactate derived from melanoma facilitates polarization of M2 macrophages, which subsequently leads to modifications in melanoma phenotypes via TCA cycle and TGF-β signaling.

Introduction

Cutaneous melanoma, a highly lethal type of skin cancer, is widespread due to the extensive distribution of melanocytes (Mei et al., 2021). Ultraviolet radiation is a predominant risk factor for cutaneous melanoma (Berge et al., 2020). The disease becomes extremely metastatic due to limited access to early detection and treatment. Although surgical resection is the primary treatment option for patients with early-stage melanoma, the likelihood of recurrence in late-stage melanoma following resection is extremely elevated (Shi et al., 2017). Despite attempts to provide more effective clinical treatments for melanoma, the prognosis for patients with advanced melanoma remains bleak. Therefore, it is crucial to elucidate the molecular mechanisms driving the development and progression of cutaneous melanoma. Currently, it is widely accepted that the etiology of melanoma involves mutated driver genes (Viraljai et al., 2021), immune checkpoints (Mahdiabadi et al., 2023), epigenetic modifications (Nanamori & Sawada, 2022) and metabolic reprogramming (Smith, Rao & McArthur, 2016) and other factors. Additionally, emerging evidence indicates that a substantial number of tumor-infiltrating immune cells, especially macrophages, are involved in pathogenesis of melanoma (Liu et al., 2021; Wu et al., 2022).

Macrophages are classified into two main types: activated M1 macrophages and alternatively activated M2 macrophages. Tumor-associated macrophages (TAMs) are main participants of the tumor microenvironment (TME). TAMs infiltration and phenotype have been associated with tumor growth (Nixon et al., 2022), angiogenesis (Li et al., 2023), invasion (Bao et al., 2022) and metastasis (Wu et al., 2020). It is worth noting that a high density of TAMs in the TME is positively correlated with poor prognosis and drug resistance in most human cancers. In melanoma, macrophages are recognized as significant contributors to tumor growth and survival. Macrophage-tumor cell interactions are associated with melanoma progression, metastasis and prognosis (Park et al., 2016; Luan et al., 2022). Exosomes have emerged as a key mechanism through which macrophages interact with malignant melanoma cells. A growing body of evidence suggests that melanoma exosomes play a crucial role in promoting tumor progression and metastasis. In addition to exosomes, melanoma cells secrete various substances, including cytokines, lactate and other small molecular elements. These components can significantly influence macrophage polarization (Bardi, Smith & Hood, 2018).

Historically, lactate has been considered a byproduct of glycolytic metabolism. But emerging evidence substantiates its significant involvement in cellular physiological processes and diseases (Moreno-Yruela et al., 2022). Lactate can be taken in and oxidized to furnish cellular energy, particularly following substantial glucose consumption (Brignardello et al., 2022). The acidic environment characterized by a pH range of 6.5–6.9, resulting from elevated lactate concentrations plays a crucial role in tumor invasion, angiogenesis, and resistance to apoptosis (Kooshki et al., 2022). Lactate exerts biological influences on cells involved in both innate and adaptive immune responses (Kooshki et al., 2022). Within the tumor microenvironment, lactate exerts an immunosuppressive function and promotes tumor development by inducing and recruiting immunosuppressive molecules and cells, particularly macrophages (Colegio et al., 2014; Noe et al., 2021). However, these findings require validation in malignant A375 melanoma, and the underlying mechanism necessitates further elucidation.

Research indicates that malignant melanoma is characterized with a lactate-induced acidic microenvironment and is infiltrated with mass macrophages. Hence, in the present study, we investigated the effect of lactate produced by melanoma on macrophage polarization. Additionally, we explored whether polarized macrophages alter melanoma phenotypes and the underlying roles of TCA cycle and TGF-β in modulation.

Materials and Methods

Cell lines and cell cultures

The human melanoma cell line A375 (Cat#CL-0014; Procell, Miami, China) was kindly provided by Procell Life Science & Technology Co., Ltd. The human melanocyte cell line PIG1 was obtained from Amrican Type Culture Collection (ATCC). A375 and PIG1 were cultured in Dulbecco’s Modified Eagle Medium (DMEM) (HyClone, Logan, UT, USA) containing 10% fetal bovine serum (FBS) (HyClone, Logan, UT, USA) and 1% penicillin/streptomycin (HyClone, Logan, UT, USA) in an incubator with 37 °C, 5% CO2. All cell lines were used within 10–50 passages.

PBMCs separation

Written informed consent was obtained from all subjects prior to their participation in the study. The study was conducted in accordance with the Declaration of Helsinki, and the protocol was approved by the Ethics Committee of Peking University Third Hospital (IRB00006761–M2022199). PBMCs were separated by density gradient centrifugation using human peripheral blood lymphocyte isolation fluid (Cat#7111011; DAKEWE, Shenzhen, China). Fresh anticoagulated whole blood was diluted with an equal volume of phosphate-buffered saline (PBS). The separation liquid was added to the centrifuge tube. The volume of separation fluid, anticoagulated undiluted whole blood, and PBS is 1:1:1. The mixed blood was then centrifuged at 800 g in a horizontal rotor for 30 min at room temperature. A thin and dense white film, known as mononuclear cells (including lymphocytes and mononuclear cells), formed between the plasma layer and the separation fluid layer. The white film layer was carefully transferred to another centrifuge tube and diluted with PBS and mixed by inverting. The mixture was centrifuged at 250 g for 10 min. Subsequently, 80 ul buffer and 20 ul CD14 microbeads were added to the cell pellet. After incubation at 4 °C for 15 min, PBMCs were separated. The separated PBMCs were prepared for the following experiments.

Conditioned medium preparation

PIG1 and A375 cells were seeded into 6 cm dishes at a density of 2,000,000 cells per dish in 6 ml DMEM and cultured for 48 h. The conditioned medium was collected (namely PIG1-CM or A375-CM) and centrifuged at 800 g for 5 min to remove detached cells and cellular debris. Processed conditioned medium (PIG1-CM or A375-CM) was mixed with RIPM medium containing 10% FBS at a ratio of 1:1 to incubate PBMCs for 48 h, and old media were replaced with fresh culture media to incubate cells for another 48 h. The above conditioned medium (termed PIG1-M-CM or A375-M-CM) was collected and centrifuged to remove cellular debris. The acquired conditioned medium was stored at −80 °C for future experiments.

Optimal oxygen level determination

Firstly, 4,000 A375 cells per well were seeded into 96-well plates and cultured in a hypoxic incubator (HERAcell 150i; Thermo Fisher Scientific, Waltham, Massachusetts, US) with oxygen concentrations of 1%, 2%, 3%, 4%, 5%, and 21% for 48 h. After removing the old media, 100 µl of DMEM and 10 µl of CCK-8 (Cat# C0038, Beyotime, Shanghai and Nantong, China) were added to each well. The plates were then incubated for an additional 2 h at 37 °C. Cell proliferation was assessed by measuring the optical density (OD) at 450 nm using a spectrophotometric plate reader.

Lactate content determination

Lactate was determined with lactate assay kit (Cat#BC2235; Solarbio, Beijing, China). Briefly, 3,000 cells were seeded into 96-well plates with 100 μL of DMEM. After incubation for 36 h, the cellular supernatant was collected, and lactate levels in PIG1-CM and A375-CM were measured according to the specifications provided by the kit The absorbances at 570 nm were recorded on a microplate reader (Thermo Fisher Scientific, Waltham, MA, USA) to quantify lactate production.

Cell viability

Trypan blue dye (0.4% Trypan Blue solution; Solarbio, Beijing, China) was used to assess cell viability according to the manufacturer’s instructions. Briefly, A375 cells and PBMCs were seeded into six-well culture plates and cultured under varying O2 levels (1%, 2%, 3%, 4%, 5%, and 21%) or with conditioned media containing 2-deoxyglucose (2-DG) or AA5 for 24 h, as per experimental requirements. After cell collection, viable and dead cells were counted under a microscope (Leica, Wetzlar, Hesse, Germany). Cell viability was calculated using the following formula: Viable cells/(Viable cells + Dead cells).

Western blot

Cells were lysed in RIPA lysis buffer (Solarbio, Beijing, China), which contained a phosphatase and protease inhibitor cocktail (Solarbio, Beijing, China). The cell lysates were centrifuged, and protein content was quantified using a bicinchoninic acid (BCA) Protein Assay kit (Beyotime, Shanghai and Nantong, China). Twenty-five micrograms of protein samples were separated by 10% sodium dodecyl sulfate-polyacrylamide gel electrophoresis (SDS-PAGE) and transferred onto polyvinylidene fluoride (PVDF) membranes with 0.45 μm pores (Millipore, MA, USA). After blocking with 5% bovine serum albumin (BSA) for 1 h at room temperature, the membranes were incubated with primary antibodies (HIF1α,1:1,000, CST; β-actin,1:3,000, CST) overnight at 4 °C, followed by three washes. Subsequently, the membranes were incubated with horseradish peroxidase (HRP)-conjugated secondary antibodies, washed with Tris-buffered saline (TBS) containing 0.1% Tween 20 (TBST). Protein bands were visualized using an enhanced chemiluminescence system (Amersham Pharmacia Biotech, Amersham, United Kingdom). Densitometric analysis was performed with ImageJ software, and relative protein levels were normalized to β-actin.

Flow cytometry

Flow cytometry was performed to evaluate M1/M2 marker expression. PBMCs were incubated in conditioned medium that contained AZD3965 (Cat#S7339; Selleck), lactate (Cat#E4473; Selleck, Washington, USA), 2-deoxy-D-glucose (2DG) (S4701; Selleck) or AA5 (Cat#ab144194; Abcam, Cambridge, USA) as experimental requirements. Cells were collected after centrifugation at 500 g for 5 min. Cell pellets were incubated with Human TruStain FcXTM (Fc Receptor Blocking Solution) (Cat#422301; Biolegend, San Diego, CA, USA) for 10 min at room temperature to block Fc receptors on human cells. To fix and permeabilize cells, they were incubated with Cyto-Fast™ Fix/Perm Solution (Cat#426803; Biolegend, San Diego, CA, USA) for 30 min at 4 °C. Cells were stained with Brilliant Violet 421™ anti-human CD206 (Cat#321126; Biolegend, San Diego, CA, USA), PE anti-human CD68 (Cat#333808; Biolegend, San Diego, CA, USA) and PerCP/Cyanine5.5 anti-human ARG1 (Cat #369710; Biolegend, San Diego, CA, USA) for 40 min at 4 °C in the dark. Fluorescence intensities of different antibody were determined with a flow cytometer (BD, Franklin Lakes, New Jersey, USA).

ECAR and OCR analysis

ECAR and OCR were analyzed using the XF96 extracellular flux analyzer (Seahorse Bioscience, North Billerica, MA, USA), following the protocol described in our previous study (Wang et al., 2021, 2023). Briefly, to begin with, PBMCs were treated with different levels of 2-Deoxy-D-glucose (2-DG) or Atpenin A5 (AA5). 4 × 104 PIG1, A375 or PBMCs underwent with different treatments were seeded into a well of seahorse cell culture plate and cultured for 24 h. The Seahorse XF stress test assay medium was prepared in accordance with the protocol for glycolysis and mitochondrial stress tests. Cells were washed three times with the conditioned medium, and the final volume in each well was 175 μl before the seahorse assay. For ECAR determination, cells were treated with sequential injections of glucose (10 mM), oligomycin (2 μM) and 2DG (50 mM). While for OCR measurement, oligomycin (2 μM), FCCP (2 μM) and rotenone/antimycin A (1 μM) were loaded sequentially. ECAR and OCR are reported as mpH/min and pmol/min, respectively.

ELISA

TNF-α and TGF-β concentrations were detected by TNF-α ELISA kit (Cat#Z10020761; CUSABIO, Houston, USA) and TGF-β ELISA kit (Cat#Z09020760; CUSABIO, Houston, USA), respectively. Samples were prepared as experimental requirements. The measurement was performed according to user manual. Briefly, 100 μl standard or sample was added to each well and incubate 2 h. The liquid was removed. A total of 100 ul Biotin-antibody (1x) was added to each well and incubate 1 h. After aspiration and wash 3 times, 100 μl HRP-avidin (1x) was added to each well and incubate 1 h. 90 μl TMB substrate was added to each well and incubate 30 min at 37 °C in the light. Finally, 50 μl stop solution was added to each well and read at 450 nm within 5 min.

Proliferation assays

Proliferation was evaluated with Cell Counting Kit-8 (CCK8) (Cat#CK04, DOJINDO, Kumamoto, Japan). Briefly, 4 × 103 cells were seeded into 96-well culture plates and cultured in conditioned medium, such as PIG1-M-CM and A375-M-CM, for 48 h. After removing cell supernatant with a pipette, 100 μl fresh DMEM stock solution and 10 μl CCK8 reagent were added to each well. After incubation at 37 °C for 30 min in the dark, the optical density (OD) values at 450 nm were recorded on a microplate reader. Three replicates of each assay were performed.

Clone formation of A375

Each well of six-well plates was seeded with 1,000 A375 cells and incubated with different conditioned media such as PIG1-M-CM, A375-M-CM etc for 2 weeks. The media were changed every 3 days and cell status was observed under microscope. Cells were washed twice with PBS and fixed with pure methanol for 15 min, followed by staining with 0.1% crystal violet (Cat#G1063; Solarbio, Beijing, China) for 15 min. The dyeing liquid was slowly washed away with running water and pictures were taken with a digital camera. The clone numbers were counted.

Migration and invasion assays of A375

Migration was evaluated using transwell chambers with 8.0 μM pores (Cat#3422; Corning, NY, USA). Briefly, A375 cells were cultured with different conditioned media. A total of 50,000 pretreated cells were seeded into the upper compartment with 200 μl DMEM, and the lower compartment was filled with 500 μl complete medium containing 10% FBS. Cells were cultured for 24 h. Migrated cells were fixed with 4% paraformaldehyde (Cat#1110; Solarbio, Beijing, China) for 15 min. Cells in the upper chambers were removed with cotton swabs. And then migrated cells were stained with 0.1% crystal violet for 15 min. Five random fields were chosen to count the number of migrated cells using a microscope (100 × magnification).

The invasion assay in our study differed slightly from the migration assay. Matrigel was diluted in DMEM at a volume ratio of 1:6. A total of 40 μl of diluted Matrigel (Cat#356234; BD) was added to the upper chamber. Subsequently, 50,000 pretreated cells were seeded into the upper chamber and cultured for 36 h. The remaining steps were performed identically to those performed in the migration assay.

Apoptosis assay

A375 apoptosis levels were detected with an Annexin V-PE/PI apoptosis kit (Cat#KGA1030; KeyGEN, Nanjing, China) following the manufacturer’s instructions. Briefly, 200,000 A375 cells were seeded into six-well plates and treated with different conditioned media for 48 h. Cells were collected using 0.25% trypsin without ethylenediaminetetraacetic acid (EDTA) and then resuspended in 50 μl binding buffer along with 5 μl propidium iodide (PI). The cell suspension was incubated in the dark for 15 min. Subsequently, 450 μl binding buffer and 1 μl Annexin V-PE were added to the suspension, which was then incubated for an additional 15 min in the dark. Finally, apoptosis levels were analyzed by flow cytometry (BD).

Statistical analysis

Statistical analyses of all data were performed with SPSS (version 26.0; SPSS Inc.). Data are shown as mean ± standard deviation. Student’s t-test was performed to analyze differences between two groups, and one-way ANOVA was used to analyze intergroup differences with a Bonferroni multiple comparison post-test. Experiment data represent at least three sample replicates. The level of statistical significance was p < 0.05.

Results

A375-CM induces macrophage M2 polarization

To investigate whether melanoma influences M1/M2 polarization, PBMCs were cultured with PIG1-CM or A375-CM. Flow cytometry results showed that the expression of the M1 macrophage surface marker CD68 (Figs. 1A, 1B) was reduced in PBMCs treated with A375-CM compared to those incubated with PIG1-CM. In contrast, the levels of the M2 macrophage marker CD206 (Figs. 1C, 1D) and arginase 1 (ARG1) (Figs. 1E, 1F) were elevated in PBMCs co-cultured with A375-CM compared to those cultured with PIG1-CM or normal medium (control group). ELISA analysis revealed that the levels of TNF-α (indicative of M1 polarization) in A375-CM did not differ from those in the control and PIG1-CM groups (Fig. 1G). However, the secretion of TGF-β (indicative of M2 polarization) from PBMCs treated with A375-CM was significantly higher than that observed in the PIG1-CM group and the control group (Fig. 1H). These findings collectively suggest that A375-CM promotes M2 polarization.

Figure 1 A375-CM induces macrophage M2 polarization.

PBMCs were co-cultured with PIG1-CM or A375-CM for 48 h. Macrophage M1 marker CD68 (A,B) and M2 marker CD206 (C,D) expressions were determined with flow cytometry. Intracellular ARG1 expression in PBMCs was measured upon co-culture with conditioned media (E,F). TNF-α (G) and TGF-β (H) levels were detected by ELISA. Data are presented as Mean ± S.D. (n = 6). *P < 0.05, ***P < 0.001 vs. control. ## P < 0.01, ### P < 0.001 vs. PIG1-CM.

Hypoxia enhances macrophage M2 polarization induced by A375-CM

A375 and PBMCs were cultured under various hypoxic conditions. Proliferation results (Fig. S1A) indicated that hypoxia (1%, 2%, and 3% O2) promoted A375 proliferation. However, cell viability assessments (Figs. S1B, S1C) revealed that 1% and 2% O2 levels significantly inhibited PBMC viability. Consequently, 3% O2 was chosen as the optimal condition for further studies. Our results demonstrated that hypoxia suppressed the expression of the M1 marker CD68 (Figs. 2A, 2B) in PBMCs treated with either PIG1-CM or A375-CM, with a more pronounced effect observed in the A375-CM group. In contrast, expressions of M2 markers, including CD206 (Figs. 2C, 2D) and arginase 1 (ARG1) (Figs. 2E, 2F), were elevated in PBMCs co-cultured with A375-CM following hypoxia treatment. In addition, cytokine secretions are also modulated by hypoxia. In PBMCs co-cultured with A375-CM under hypoxic conditions, TNF-α levels (Fig. 2G) remained unchanged, while TGF-β levels (Fig. 2H) increased. Collectively, these findings suggest that hypoxia reinforces M2 polarization induced by A375-CM.

Figure 2 Hypoxia enhances macrophage M2 polarization induced by A375-CM.

PBMCs co-cultured with A375-CM were treated under hypoxic condition (3%) for 48 h. CD68 (A,B), CD206 (C,D), and ARG1 (E,F) expressions were detected using flow cytometry. After hypoxic treatment TNF-α (G) and TGF-β (H) levels in conditioned media were measured by ELISA. Data are presented as mean ± S.D. (n = 6). *P < 0.05, ***P < 0.001.

M2 polarization of macrophage induced by A375-CM is regulated by lactate

Firstly, optimal concentrations of lactate and AZD3965 were determined by evaluating CD206 expression in PBMCs (Figs. S2A–S2D). The lactate production from A375 was found to be higher than that from PIG1 (Fig. S2E). To explore whether lactate in A375-CM alters macrophage polarization, the MCT1 inhibitor AZD3965 was utilized to block lactate transport into cells. In this study, flow cytometric analysis revealed that CD68 expression in PBMCs co-cultured with control medium or PIG1-CM decreased upon the addition of lactate (Figs. 3A, 3B). In contrast, CD68 expression in PBMCs co-cultured with A375-CM increased following treatment with AZD3965 (Figs. 3A, 3B). Furthermore, expressions of CD206 (Figs. 3C, 3D) and ARG1 (Figs. 3E, 3F) in PBMCs exhibited opposite trends. Their expressions increased in the control or PIG1-CM groups after lactate treatment, but decreased when cells cultured with A375-CM were treated with AZD3965. While TNF-α secretion was unaffected by treatment with lactate or the MCT1 inhibitor (Fig. 3G), TGF-β secretion (Fig. 3H) from PBMCs increased upon the addition of lactate to PIG1-CM, whereas it decreased following the addition of AZD3965 to A375-CM.

Figure 3 M2 polarization of macrophage induced by A375-CM is regulated by lactate.

The lactic acid concentration in A375-CM and PIG1-CM is measured by spectrophotometry with lactate kit (A). PMBCs co-cultured with conditioned media were treated with MCT1 inhibitor AZD3965 (Cat#S7339, Selleck, USA) for 48 h to block lactate transported in cells. M1 marker CD68 (B,C), M2 markers CD206 (D,E) and ARG1 (F,G) expressions in PBMCs were detected after inhibition of lactate transport. TNF-α (H) and TGF-β (I) levels secreted from polarized macrophages were measured using ELISA. Data are presented as Mean ± S.D. (n = 6). **P < 0.01, ***P < 0.001.

TCA cycle instead of glycolysis is in involved in macrophage M2 polarization induced by lactate

Melanoma is characterized by the Warburg effect. The glycolysis and mitochondrial respiration of A375 cells were analyzed using a Seahorse analyzer. The results indicated that ECAR of A375 was higher than that of PIG1 (Fig. 4A), while OCR of A375 was lower compared to PIG1 (Fig. 4B). To assess the inhibitory effects on PBMCs, the glycolysis inhibitor 2-DG and the mitochondrial respiration inhibitor AA5 were utilized to evaluate their impacts on ECAR (Fig. 4C) and OCR (Fig. 4D), respectively. Concentrations of 1 mM for 2-DG and 1 μM for AA5 were employed for further studies. Additionally, we blocked glycolysis and the TCA cycle to analyze their roles in lactate-induced M2 polarization. Our findings showed that glycolysis inhibition did not affect the expression of CD68, CD206 and ARG1, nor did it alter TGF-β and TNF-α secretion. In brief, glycolysis did not influence macrophage polarization induced by A375-CM. Interestingly, blockage of TCA cycle enhanced the expression of M1 markers, such as CD68 (Figs. 4E, 4F), while inhibiting the expression of M2 markers (CD206 and ARG1) (Figs. 4G–4J). Furthermore, the results of cytokine TGF-β (Fig. 4L) supported these conclusions. Collectively, these results demonstrate that lactate induces M2 polarization via TCA cycle.

Figure 4 TCA instead of glycolysis is in involved in macrophage M2 polarization induced by lactate.

PBMCs were induced to M0 Macrophage after being treated with 100 ng/ml PMA for 48 h. And then cells were treated with lactate, lactate and 2DG or lactate and AA5 for another 48 h, respectively. Macrophage markers (CD68, CD206 and ARG1) (A–F) expressions were determined with flow cytometry, and cytokine secretion (TNF-α and TGF-β) (G, H) were investigated by ELISA. Data are presented as Mean ± S.D. (n = 6). *P < 0.05, **P < 0.01, ***P < 0.001 vs. lactate group. # P < 0.05, ##P < 0.01, ### P < 0.001 vs. lactate+2DG group.

M2 macrophages promote melanoma proliferation, cell cycle progression, clone formation, invasion and inhibit apoptosis

To investigate the impact of polarized macrophages on A375 phenotypes, PIG1-M-CM and A375-M-CM were first prepared to culture A375 cells. CCK8 assays indicated that A375-M-CM significantly promoted the proliferation of A375 compared to PIG1-M-CM (Fig. 5A). Consistently, A375-M-CM markedly promoted A375 clone formation (Figs. 5B, 5C) and increased the cell ratio in the G2/M phase (Figs. 5D, 5E). These findings suggest that polarized macrophages induced by A375-CM promote A375 growth. Additionally, flow cytometry analysis demonstrated that the apoptotic level of A375 cells was inhibited following treatment with A375-M-CM (Figs. 5F, 5G). Transwell assays revealed that co-culturing A375 cells with A375-M-CM resulted in enhanced migration (Figs. 5H, 5I) and invasion (Figs. 5J, 5K).

Figure 5 M2 macrophages promote melanoma proliferation, cell cycle, clone formation, invasion and inhibit apoptosis.

Conditioned medium of PBMCs that was co-cultured with PIG1-CM or A375-CM was collected and was utilized to incubate A375. The proliferative effect of polarized macrophages on A375 was observed with CCK8 kit (A). Clone formation (B,C) and cell cycle (D,E) were also performed to assess proliferation. Apoptosis was evaluated by flow cytometry using the Annexin V-PE/PIApoptosiskit (F,G). Migration (H,I) and invasion (J,K) were investigated with transwell. Data are presented as Mean ± S.D. (n = 3). *P < 0.05, **P < 0.01, ***P < 0.001 vs. control. # P < 0.05, ### P < 0.001 vs. PIG1-M-CM.

Altered melanoma phenotypes by M2 macrophages are modulated by TCA and TGF-β secretion

Our findings indicate that polarized macrophages have an impact on various melanoma phenotypes. However, the specific mechanism underlying these alterations requires further investigation. Herein, we explored whether TCA cycle exerts a role in melanoma phenotype alterations induced by polarized macrophages. We observed that A375-M-TCA inhibitor-CM decreased A375 proliferation, as detected by CCK8 test (Fig. 6A). In addition, TCA cycle inhibitor of macrophages inhibited A375 clone formation (Figs. 6B, 6C) and cell cycle progression (Figs. 6D, 6E), and promoted apoptosis (Figs. 6F, 6G). Moreover, the migration (Figs. 6H, 6I) and invasion (Figs. 6J, 6K) of A375 were attenuated upon TCA cycle inhibition in polarized macrophages. Despite these observed alterations in melanoma phenotypes the specific mediators involved in the interaction between these two cell types remain unclear. Our above findings showed TCA cycle enhances macrophage M2 polarization induced by lactate and promoted TGF-β secretion. Therefore, we hypothesize that the interaction between polarized macrophages and melanoma cells is mediated through TGF-β. Notably, the elimination of TGF-β from A375-M-CM through the use of neutralizing antibodies resulted in reduced A375 proliferation (Fig. 6A). The similar results were observed in clone formation (Figs. 6B, 6C) and cell cycle progression (Figs. 6D, 6E), migration (Figs. 6H, 6I) and invasion (Figs. 6J, 6K). In contrast, the level of apoptosis in A375 cells increased following the removal of TGF-β from the conditioned medium.

Figure 6 Altered melanoma phenotypes by M2 macrophages are modulated by TCA and TGF-β secretion.

Polarized macrophages were treated with TCA inhibitor AA5 and the conditioned medium was collected to culture A375. Additionally, TGF-β was removed from A375-M-CM with TGF-β neutralizing antibody and then the conditioned medium was used to incubate A375. CCK8 (A), clone formation (B,C) and cell cycle (D,E) were performed to evaluate cell proliferation. Apoptosis was assessed by flow cytometry (F,G). Migration (H,I) and invasion (J,K) were evaluated with transwell. Data are presented as Mean ± S.D. (n = 3). *P < 0.05, **P < 0.01, ***P < 0.001 vs. A375-M-CM, ##P < 0.01 vs. A375-M-TCA inhibitor-CM.

Discussion

Macrophages play a crucial role in the pathological mechanism of human melanoma which is characterized by the Warburg effect and elevated lactate levels. However, whether melanoma high lactate environment affects macrophage polarization and the underlying mechanisms need further investigation. In the present study, we observed that melanoma promoted M2 macrophage polarization, as evidenced by increased expressions of CD206 and ARG1, as well as secretion of TGF-β. Conversely, M1 polarization was inhibited, as indicated by decreased CD68 expressions. Interestingly, hypoxia and lactate were found to enhance the M2 polarization process. Inhibition of lactate transport into cells resulted in the attenuation of M2 polarization induced by A375-CM. Notably, melanoma-derived lactate was found to regulate M2 polarization through TCA cycle rather than glycolysis. Furthermore, polarized macrophages influenced various melanoma phenotypes, such as proliferation, clone formation, cell cycle, apoptosis, migration, and invasion through TCA cycle enhance and TGF-β secretion.

It has been demonstrated that tumor cells interact with TAMs. In the present study, we detected that melanoma affected macrophage polarization. PBMCs were cultured with A375-CM, and results showed that expressions of M2 markers CD206 and ARG1 in PBMCs treated with A375-CM were higher compared to those in normal melanocyte group. CD68 expression in A375-CM group decreased. Additionally, TGF-β level in A375-CM group enhanced. M0 macrophages do not secrete TNF-α, and treatment with A375-CM inhibited M1 polarization. Consequently, M0 macrophages treated with A375-CM did not secrete TNF-α, resulting in no significant difference in TNF-α levels between the control and A375-CM groups. Despite the lack of change in TNF-α levels, we can conclude that melanoma inhibits M1 polarization based on the CD68 expression results. Interactions between melanoma and macrophages have been reported in other studies. Bardi, Smith & Hood (2018) reported that melanoma exosomes (B16-F10 and RAW264.7) promote a mixed M1 and M2 phenotype in macrophages. Additionally, altered macrophage polarization has the potential to impede the progression of melanoma (Cao et al., 2019).

Pathologic hypoxia is prevalent in various solid tumors (Pan et al., 2017; Wang et al., 2021) including melanoma (Liu et al., 2019) and is considered a main hallmark of tumor microenvironment. In our study, we evaluated the effect of hypoxic conditions on M1/M2 polarization induced by A375-CM. Our results indicated that hypoxia promoted the expression of M2 markers while suppressing M1 marker expression. These outcomes align with previous studies, such as the research conducted by Zhang et al. (2022) which demonstrated that hypoxia induces the M2 phenotype to facilitate cancer aggressiveness in glioblastoma. Xu et al. (2021) demonstrated that under hypoxia, glioma-derived exosomes promoted M2-like macrophage polarization by enhancing autophagy. Additionally, hypoxia-induced tumor exosomes promoted M2 macrophage polarization in infiltrating myeloid cells (Park et al., 2019).

Lactate, once considered merely a byproduct of glycolysis, is now recognized for its significant role in cellular physiology and pathogenesis of diseases. It serves as an alternative energy source for cells, particularly under condition of glucose depletion. Lactate-lactylation is implicated in tumor metabolic reprogramming and tumor immunity (Chen et al., 2022). However, the role of lactate in modulation of macrophage polarization needs further investigation. In the present study, we investigated whether lactate derived from melanoma regulated M1/M2 polarization. Our findings indicated that melanoma A375 produced a higher level of lactate compared to normal melanocyte PIG1, indicating that melanoma is featured with Warburg effect and mass lactate production. Notably, when the lactate shuttle into cells was obstructed using an MCT1 inhibitor, we observed a discernible alteration in macrophage polarization in macrophages cultured in A375-CM. The MCT1 inhibitor mitigated M2 polarization induced by A375-CM and enhanced M1 polarization. Recent studies have elucidated the association between lactate and macrophage polarization. Zhang et al. (2020) reported that endothelial derived lactate induced M2-like macrophage polarization, thereby facilitating muscle regeneration following ischemia. Professor Yang found that lactate suppressed M1 polarization by inhibition of YAP and NF-kB activation (Yang et al., 2020).

TCA cycle not only contributes to macromolecule synthesis, but also plays a critical role in chromatin modification, post-translational modifications and immune response (Martinez-Reyes & Chandel, 2020). Metabolites in TCA cycle, such as citrate, itaconate, succinate, and fumarate, function in extensive immunoregulation. It has been revealed that lactate can be absorbed and oxidized by TCA cycle under various conditions. We speculated that TCA cycle may exert an essential role in macrophage polarization induced by lactate. In our experiments, macrophages cultured in media containing lactate were treated with a TCA cycle inhibitor or a glycolysis inhibitor. The results showed that TCA cycle inhibitor decreased the expression of M2 markers, including CD206, ARG1 and TGF-β, while increasing M1 marker CD68 expression. The above results indicated that TCA cycle promotes M2 polarization. However, glycolysis inhibitor did not affect M2 polarization, in that M2 macrophages are more dependent on oxidative phosphorylation (OXPHOS) (Viola et al., 2019). Above all, glucose metabolism plays a crucial role in macrophage polarization, with glycolysis and TCA cycle promoting M1 and M2 polarization, respectively (Mouton et al., 2020; Viola et al., 2019). Additionally, miR-223 has been shown to interfere with the glycolysis pathway through the downregulation of HIF-1α, resulting in an anti-inflammatory macrophage phenotype (Dang & Leelahavanichkul, 2020). Kim et al. (2022) reported that Cassiaside C dampened M1 polarization by glycolysis downregulation. Trauelsen et al. (2021) found extracellular succinate hyperpolarized M2 macrophages through SUCNR1/GPR91-mediated Gq signaling.

Cancer cells engage in interactions with immune cells in tumor microenvironment. In our study, we found that malignant melanoma A375 induced macrophage polarization. We further investigated the impact of polarized macrophages on melanoma phenotypes. We observed that M2 macrophages promoted A375 proliferation, cell cycle progression, clone formation, migration and invasion, while inhibited apoptosis. The underlying mechanism was also further clarified. Our results revealed that TCA cycle in polarized macrophages contributes to the alterations in melanoma phenotypes. However, the specific elements that mediate the influence of macrophages on melanoma remain unclear. TGF-β, a common type of cytokine, is secreted by various cell types, such as macrophages, epithelial cells, endothelial cells, neutrophils and fibroblasts (Resende et al., 2022; Cheng et al., 2022, 2018; Tu et al., 2023). Increasing studies have demonstrated that TGF-β exerts a key role in vascular remodeling, proliferation and cellular tissue invasion and related mechanisms have been elucidated in different diseases (Cheng et al., 2016). Interestingly, our findings demonstrated that cytokine TGF-β secreted by macrophages altered melanoma phenotypes.

It is important to distinguish between macrophage-conditioned medium and the macrophages themselves. However, achieving the experimental conditions needed to directly observe interactions between these two cell types can be challenging. An alternative approach is to use conditioned culture medium to co-culture other cells, which can help fulfill the objectives of the study.

Additionally, firstly, it is necessary to replicate these findings in additional melanoma cells obtained from melanoma patients. Secondly, the specific mechanism by which lactate influences macrophage polarization by regulating TCA cycle remains unclear. Lastly, further investigations are needed to explore the impact of lactate-induced macrophage polarization on melanoma cells.

Conclusions

In conclusion, lactate derived from melanoma promotes M2 macrophage polarization and which in turn alters melanoma phenotypes via TCA cycle activation and TGF-β secretion. Our study may provide new insights into pathogenesis of melanoma from perspectives based on Warburg effect and immune mechanisms.

Supplemental Information

Supplemental Information 1 The optimal O2 determination.

(A) A375 proliferation under different hypoxic O2 levels. (B-C) A375 cell viability and PBMCs cell viability under hypoxic O2 levels. (D) Western Blot of HIF1α in PBMCs under 3% O2. ***P<0.001 versus 21%, ###P<0.001 verse 3% n=3.

Supplemental Information 2 Lactate and AZD3965 levels determination.

(A-B) The impact of lactate on CD206 expression in PBMCs. (C-D) The influence of AZD3965 on CD206 expression in PBMCs. (E) Lactate levels in PIG1 and A375 conditioned media. *P< 0.05, **P<0.01, ***P<0.001 versus control, n=3.

Supplemental Information 3 2DG and AA5 levels determination.

(A-B) 2DG and AA5 levels were determined via PBMC viability. **P<0.01, ***P<0.001 versus 0, n=3.

Supplemental Information 4 Raw data of Figure 1-6 for statistical analysis.

We are extremely grateful for the plastic surgeon and the central laboratory platform for the specimens and services, respectively.

Additional Information and Declarations

Competing Interests

The authors declare that they have no competing interests.

Author Contributions

Qifei Wang conceived and designed the experiments, performed the experiments, prepared figures and/or tables, authored or reviewed drafts of the article, and approved the final draft.

Yurui Shi conceived and designed the experiments, performed the experiments, prepared figures and/or tables, and approved the final draft.

Zelian Qin conceived and designed the experiments, prepared figures and/or tables, and approved the final draft.

Mengli Xu performed the experiments, prepared figures and/or tables, and approved the final draft.

Jingyi Wang analyzed the data, prepared figures and/or tables, and approved the final draft.

Yuhao Lu performed the experiments, analyzed the data, prepared figures and/or tables, authored or reviewed drafts of the article, and approved the final draft.

Zhenmin Zhao conceived and designed the experiments, authored or reviewed drafts of the article, and approved the final draft.

Hongsen Bi conceived and designed the experiments, authored or reviewed drafts of the article, and approved the final draft.

Human Ethics

The following information was supplied relating to ethical approvals (i.e., approving body and any reference numbers):

Peking University Third Hospital Research Ethics Board

Data Availability

The following information was supplied regarding data availability:

The data are available in the Supplemental Files.

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
