# Peer review of "A375 melanoma-derived lactate controls A375 melanoma phenotypes by inducing macrophage M2 polarization via TCA cycle and TGF-β signaling"

_PeerJ, doi:10.7717/peerj.18887_

## Round 0.1 · original submission · Major Revisions

· Academic Editor

Major Revisions

Please provide a detailed rebuttal letter to all of the reviewers.

**Language Note:** The review process has identified that the English language must be improved. PeerJ can provide language editing services - please contact us at [email protected] for pricing (be sure to provide your manuscript number and title). Alternatively, you should make your own arrangements to improve the language quality and provide details in your response letter. – PeerJ Staff

Reviewer 1 ·

Basic reporting

- The authors must review and correct the English language of the whole manuscript as it contains grammatical, syntax, and spelling errors.
- The past tense and passive voice writing are thoroughly recommended in a scientific manuscript when describing actions already done.
- Lane 97. All abbreviations need to be fully described at the first mention, such as peripheral blood mononuclear cells (PBMCs), mainly in a title.
- Please avoid imprecision in words such as "certain" or "different mediums."
-A blank space between numbers and units is required (250 g or 20 mL); please correct this throughout the manuscript.
-A sentence should never start with a number.
- Minutes or min should be consistently used in the same form throughout the manuscript.
-Lactic acid is not a noun that needs to be capitalized; the same goes for macrophage polarization (lane 326) and citrate in lane 332.
-Temperature should always be expressed in Celsius degrees, and units indicated as °C; please correct.
-Centrifugation units must be homogenous in the manuscript; either rpm or gravities are correct, but always the same units; please correct.
- A sentence should never start with an And, please correct the manuscript and figure legends.
-In the results section, sentences from lanes 193-195, 209-213, 221- 224, 235- 239, 248-252, and 254-256.…. are not results descriptions and should included in the methods section to explain what for each assay was conducted.
-Figure 3-small graphics need to be re-aligned to be easy to follow.
A/ B-C/ D-E/ F-G/ H-I.
-Figure 4 has many mistakes in “lactate.” Please correct them.
-The correct use of commas, dots, and semicolons should be corrected along with methods to make the text clear.
-Lane 332, correct aconitate
-The marker CD163 (lanes 296-297) was not analyzed in this manuscript; please correct this.

Experimental design

- The methods section needs to be profoundly improved since it is unclear and lacks information. There needs to be a better explanation about the why? and how? of each section.
- As an example, the word hypoxia is first mentioned in the results section (lane 208) to describe the effect of hypoxia. However, there is no previous mention of this in methods, so it is difficult to follow the methods section, which needs clear descriptions and their related results.
- - How many cultured PIG1 and A375 cells replicates were seeded at each assay?; how many replicates for each assay or experiment were conducted to validate results statistically?.
- - Lactate content determination is a central method in this manuscript but is scarcely described; please include a brief description of the method, including the specific data needed to be a reproducible method and the number of replicates.
- The flow cytometry, ELISA assay, proliferation assay, clone formation, migration and invasion assay, and apoptosis assay texts are written in different styles. Some are described as a protocol to be followed in the present tense, and others are correctly described in passive voice and past tense; please homogenize and correct, including the missing information, to allow reproducibility of each technique. Please include a description of the various abbreviations to make the text clear and easy to understand.
- Flow cytometry, what does Fc receptors mean? Please include the concentrations used for each stain and the number of replicates.
- Proliferation assays. “certain” conditioned medium is unacceptable; please include the medium composition. Explain how the cell supernatant was removed. How many replicates from how many samples were determined? How do the cells were centrifuged before removing supernatants? Which DMEM concentration do they use?
- Clone formation. This subtitle needs to be clarified: clones of what? Please clearly describe this assay.
- Please include a clear and deep description of “different conditioned mediums.” It is difficult to understand how this procedure was done; please rewrite the paragraph as it contains many grammatical mistakes and explain enough to be reproducible.
- Migration of what?… and invasion of what? Please complete subtitles and include subjects in the sentences.
- Most methods sections need more information. Thus, it is difficult to understand what they do and how they do it. Please complete the information.

Validity of the findings

- The results section should only include the description of the results. Findings and interpretations of results should be included and clearly described in the discussion section.
- From figures 1A-H it is easy to conclude the induction of macrophage M2 polarization; however, since CD68 is not a specific marker for M1 macrophages, it is sometimes highly expressed in M2 macrophages, the low or not induced macrophages M1polarization should be analyzed using a marker as CD86 which is specific for M1 macrophages.
- The results Section (lanes 246-259) explains what the authors made, but there needs to be a precise analysis and description of the resulting data included.
- The melanoma effect on M1 macrophage polarization should be revised; some results are not conclusive since CD68 is not an M1-macrophages-specific marker. There is a difference between inhibiting and not-inducing; this has to be reanalyzed since TNF-a results often did not significantly change (Figs 1-3).
- The authors should better explain why macrophage M1 polarization results show statistical differences when CD68 is determined but no differences when TNF-a levels are determined. It needs to be made clear how M1 macrophage polarization is induced.
- The role of TCA in the induction of a shift from M1 to M2 macrophages (lanes 339) is disputable if not based on additional evidence. The TCA induces the polarization of macrophages to the M2 phenotype, but a shift from one phenotype to another should be carefully stated.
- The idea in lanes 339-343 needs to be better explained, and the claims need to be supported by confirmatory literature to be conclusive.

Reviewer 2 ·

Basic reporting

In the manuscript titled “Melanoma-derived lactate controls melanoma phenotypes by inducing macrophage M2 polarization via TCA cycle and TGF-b signaling” by Wang et al. the authors aim to “investigate the effect of lactate produced by melanoma on macrophage polarization and explore whether polarized macrophages alter melanoma phenotypes and the underlying role of TCA cycle and TGF-B modulation” As reviewer, I find interesting the effort of the authors to use drugs that modify tumor-metabolic reprogramming of innate cells in melanoma progression. However, this reviewer finds that the data is still preliminary, as it only partially answers their proposed aims, particularly the role of metabolism in macrophage polarization. Melanoma is a highly heterogeneous disease with widely recognized traits. One of those well-known traits is the author’s claim that tumor cells or lactate induce polarization of macrophages towards the M2 phenotype within tumors. The latter is a particularly well-known feature of the widely used melanoma cell line A375. In fact, such trait is universal to tumors. Hence, their use of only 1 melanoma cell line questions the novelty of their findings. By contrast, this reviewer considers a miss opportunity that the authors haven’t much compared the metabolic profiles of A375 cells to that of the transformed melanocyte cell line PIG1. Minor point for basic reporting: The title of the paper is quite odd, don’t know what the authors refer as “melanoma phenotypes”. They need to revise this title. Authors must correct different typos and grammatical errors throughout their submission. The data on figures 5D and 6D are impossible to evaluate.

Experimental design

There are several errors in their experimental design. As indicated above their aims are quite wide and ambitious, except for the role of TGFb. However, they need to correct other errors:
2. The PIG1 melanocyte cell line is a transformed cell line with HPV 16 and E7 genes (Le Poole et al., 1997). Although this cell line proliferates, it remains dependent on the addition of tetradecanoyl phorbol 13-acetate (TPA). The latter is known to modulate macrophage polarization (Chuthaputti et al., 1989; Welgus et al., 1986). Despite this detail, the authors reported that they cultivated both PIG1 and A375 cell lines in 10% FBS DMEM, which may alter the pigmented phenotype and dynamics of the PIG1 cells. In fact, the authors used the PIG1 cell line for proliferation and invasion assays, which clearly are bias towards mutant-driven highly proliferative cells, like A375. As a result, this reviewer concluded that more melanoma cell lines are needed to correctly interpret the outcome of these studies.
3. The data exhibit a complete lack of experimental detail and appropriate controls. For example, the conditions of the “hypoxia treatment” are not clearly stated in the material and methods section. Only a brief description appears written on the legend of figure 2, where it says that cells were maintained at 3% hypoxic condition for 48 hrs. Evans et al. (2006), have provided a quantitative evaluation of skin oxygenation showing that that in the hair follicles the oxygen tension ranges between 2.5 and 0.1% O2. Melanocytes, which reside in the dermal-epidermal junction in humans and in certain transgenic mouse lines, and in hair follicles in mice, are, therefore, in a hypoxic environment. Hence, the 3% hypoxic conditions are not the correct condition for melanoma cells, instead the authors should have use 1% O2. In any case, the authors need to provide full detail on how they achieved hypoxia and evidence that such conditions are not affecting the viability of the primary cells (PBMC) co-cultured with A375 cells, as indicated in the legend of figure 2. They also need to provide evidence that the hypoxia conditions work, such as indicate induction of Hif-1 expression or any preferred hypoxic marker. Moreover, the authors need to include A375 cells without conditioned media in the experimental bars on Figure 2 to assess the effect of hypoxia on parental A375 cells. Please also specify what are the “control” cells: Is it A375 cells without conditioned media or cells under non-hypoxic conditions.
4. The authors need to provide full detail of the use of the MCT1 inhibitor AZD3965. It is not listed in the material and methods section. Moreover, the authors indicate that PBMCs were in culture when A375 cells were treated with MCT1 (line 227). Is this correct? This is different of what was stated in the material and method section, where cells were treated only with the conditioned media. As indicated above, authors need to include the effect of MCT1 over parental A375 cells.
5. The metabolic data need extensive clarification. Metabolic reprogramming in M1-polarized macrophages features an increase in glycolytic metabolism and a decrease in mitochondrial oxidative phosphorylation (OXPHOS), as well as truncation of the tricarboxylic acid (TCA) cycle and accumulation of pro-inflammatory intermediates such as succinate. A recent study found that HIF1α-dependent glycolysis is associated with M2 macrophage differentiation, indicating that glycolysis is also essential to the M2 macrophages polarization (Yu et al., 2020). Hence, the authors need to clarify their findings and do new test the role of mitochondria in their outcomes, like using seahorse mito Stress test kits. The flow data on figure 4 needs to show individual data, not the aggregate solid bars.

6. Need to revise all material and methods for accuracy and specific description of experimental interventions.
7. Need to add normalization with cell numbers for experiments in all figures, particularly in figure 3 and 4.
8. All bars must show individual values of their 3 repeated experiments, as indicated in the Statistical analysis section.

Validity of the findings

As indicated above, the use of 1 melanoma cell line is not sufficient to validate their data. The conclusions are not new, except for the role of TGFb. However, the role of TGFb has not been sufficiently explored as it appears separated from other parts of the report. For example, what is the effect of TGFb over metabolism of melanoma cells and macrophages?

Reviewer 3 ·

Basic reporting

This manuscript would benefit from substantial revision in the following areas:
1. In line 21, “melanoma phenotypes responses” should be “melanoma phenotype responses”
2. In line 28, the sentence can be clearer if the authors revise it as below: M2 markers CD206 and ARG1 expression increased, as did TGF-β secretion. Conversely, M1 marker CD68 expression decreased.
3. In line 30, the sentence “Lactate level in A375-CM was higher than that in PIG1 condition medium (PIG1-CM).” disrupts the fluency of the writing and makes it confusing. Removing this sentence could aid the flow.
4. In line 37, lactic acid is not equivalent to lactate. Lactate but not lactic acid is produced by glucose metabolism. The authors need to revise the usage of these two items in the manuscript.
5. In line 43, “Ultraviolet radiation is established a predominant risk factor of cutaneous melanoma” is grammatically incorrect. It’s unclear what the authors wanted to express in this sentence.
6. Line 80-82, this is overstated. Lactate’s role in M2 polarization has been studied. Please refer to papers: PMIDs: 34767443 and 37564659.
7. In the section of A375-CM induces macrophage M2 polarization, Figs 1A-B should be described first. Similarly, Fig 1G should be described before 1H. Messed-up description sequences also happened in other figures, making the current flow very confusing.

Experimental design

1. It’s unclear why PIG-CM is used as a control. Have any previous findings shown this cell line does not induce M2 polarization?
2. It is known that the lactate effect is mediated by Hypoxia Inducible Factor1- a (HIF1-a). Therefore, to strengthen the conclusion in the section of Hypoxia enhances macrophage M2 polarization induced by A375-CM, the authors would need to determine if the observed effects depend on HIF1-a.
3. To demonstrate the lactate-induced M2 polarization in the section of M2 polarization of macrophage induced by A375-CM is regulated by lactate, it is important to test if adding lactate in normal media and PIG-CM conditioned media can also induce M2 polarization.
4. In line 244, the conclusion is incorrect. The result of TCA inhibition indicates TCA cycle is required for the observed lactate-induced M2 polarization. There is no evidence showing lactate-induced M2 polarization goes through the TCA cycle.
5. In sections of M2 macrophages promote melanoma proliferation, cell cycle progression, clone formation, invasion and inhibit apoptosis and Altered melanoma phenotypes by M2 macrophages are modulated by TCA and TGF-beta secretion, did authors cocultured M2 macrophage with melanoma cells? The lack of the experiment procedures makes it very hard to justify the conclusions.

Validity of the findings

It would strengthen the findings if the authors could provide all the data points in the figures. For example, the authors could consider using a scatter plot with bar.

Additional comments

No

·

Basic reporting

The current article is written in plain English and overall easy to follow, although further polishing of certain phrases would definitely add to the readability of this work.

It seems current article did not reference one of the landmark paper published by Ruslan Medzhitov group “Functional polarization of tumour-associated macrophages by tumour-derived lactic acid” (Nature, 2014. doi: 10.1038/nature13490 ) which clearly shows that lactate produced from tumor can induce M2 polarization of tumor infiltrating macrophages.

Experimental design

The current article has clearly stated and well defined research questions, which aims to fill the gap of our current knowledge. The key questions posed by the authors have the potential to enhance the clinical outcome of patients with melanoma, if properly answered.

However, the overall experimental design failed to reach the general standard of the field, lacking in several key aspects. The authors failed to conduct the most simple, and basic experiments for a lot of their results. Some of the experiments lack proper control, and authors seem to choose convoluted and indirect approaches in lieu of more simple and direct experimental design to prove their hypothesis. I will discuss the issues with experimental design and suggested improvements in more detail at section 4. Additional comments.


Methods are well described and easy to replicate

Validity of the findings

It seems that current article is somewhat redundant to previously published paper (Nature, 2014. doi: 10.1038/nature13490) and thus adds little to the body of scientific knowledge as a whole.

Additional comments

Overall, the current article relies on poorly designed experiments, and because of that, results obtained from experiments limit the validity of the author’s claim.

Major Comments:

The most notable design mistake in terms of experiments, is that throughout the entire article, the authors failed to show the direct effect of metabolite lactate in macrophage polarization. Instead of using conditioned media, the authors should directly add lactate to PBMC and show its effect on polarization.

It is the most simple experiment which can generate clear answers. Although the authors tried using the MCT1 inhibitor (AZD3965), it is still indirect evidence because MCT1 transporter can transport numerous metabolites and is not limited to only lactate.

Use of conditioned media largely hinders author's interpretation of the data, which states that the effects shown in current study is due to lactate. Especially when the authors clearly state in the introduction section, that exosomes and cytokines secreted by cancer cells can affect macrophage polarization. Conditioned media will contain exosomes and cytokines. No effort is put in place to rule out the effect of exosomes and cytokines while using conditioned media.

In Fig.1 G and H, the authors measured TNF-a and TGF-b levels from the media after culturing PBMCs with control, PIG1-CM, and A375-CM. However, it is well known that cancer cells themselves can express and secrete TGF-b. The authors should first measure the TNF-a and TGF-b levels from PIG-CM and A375-CM, and show that their level is unchanged compared to control media in order to conclude that TGF-b is derived from PBMCs.

In Fig.2, the authors defined 3% oxygen as a hypoxic condition. However, most published papers use 0.5% or 1% oxygen as a hypoxic condition. 3% oxygen, while still lower than 21% (normoxia), might not trigger HIF1a stabilization and its downstream signaling pathways which is a hallmark of hypoxic response. The authors need to show HIF1a western blot to justify using 3% oxygen as a hypoxic condition.

Also in Fig.2, there is no proper control for hypoxia. The authors should first try to compare Normoxia vs Hypoxia in order to show the effect of hypoxia on PBMC polarization. Adding hypoxia on top of A375-CM is not an ideal experimental design. Please show PBMC polarization comparing normoxia vs hypoxia using control culture conditions (RPMI + 10% serum).

In Fig.3, as mentioned before, the authors should conduct experiments by adding lactate directly into culture media, rather than using MCT1 inhibitor to conclude the effect on polarization of conditioned media is due to lactate. Added benefit of using lactate is that we can control how much lactate is added to PBMCs, and try several different concentrations of lactate in order to get the dose-response curve. If changes in PBMC polarization occurs at clinically relevant lactate concentration (the lactate level we find in tumor microenvironment of melanoma patients), it could further strengthen the relevance of current study.

In Fig.4, the authors used 2DG and AA5 as inhibitors of glycolysis and TCA, respectively. However, the concentrations of 2DG and AA5 used is not disclosed in the article (Not written in methods or Figure legend). As a general rule, different cell types have different sensitivity to chemical inhibitors. And therefore, there is no way to prove that the treatment of 2DG or AA5 (whichever concentration used by the author) actually resulted in inhibition of glycolysis or TCA cycle unless the authors show evidence of it. As a simple control experiment, the authors should test different doses of 2DG and AA5 and show dose-dependent inhibition of glycolysis and TCA cycle. Then choose appropriate concentration of inhibitors for further experiment.

In Fig.5, the authors state that “conditioned medium of PBMCs that was co-cultured with PIG1-CM or A375-CM was collected and was utilized to incubate A375”. So in essence, the authors used macrophage conditioned medium, not macrophage itself. But in the text, the authors state the effect of “polarized macrophages” on A375 cells. Data shown in Fig.5 is the effect of macrophage conditioned medium, not macrophage itself. Distinctions between the two should be clearly stated in order to avoid confusions to the readers.

Moreover, using PBMC conditioned media, which was generated using A375-CM, really confounds the interpretation of the data. The authors have no idea whether the results shown is due to something secreted from macrophages, or is it due to something that was already in A375-CM. Conditioned media on top of conditioned media is a very poor experimental design.

In Fig.6, the authors treated polarized macrophage with AA5 and harvested media to make conditioned media. AA5 is a chemical, so it means conditioned media still contain AA5 unless the authors somehow removed this chemical from the conditioned media. So essentially, “A375-M-TCA-inhibitor-CM” is basically treating cells with AA5, which would definitely damage cell proliferation. The authors have no way to distinguish whether the results observed are due to residual AA5 contained in the CM or not.

Furthermore, there is no proper control for TGF-b antibody treatment. The authors should include TGF-b antibody only treated groups for fair comparison.

Also in Fig.6, it is lot easier and straightforward to treat cells with TGF-b rather than using TGF-b antibody.



Minor comments:

Throughout the paper, FACS plots seem to use different x-axis scales, which may confuse readers. (Fig.1C A375-CM plot uses different x-axis scales from the other two plots) Please use the same scale x-axis.

In Fig.3, although panel C says CD68 is increased in the AZD-3965 treated group, looking at FACS plot in panel B seem to suggest otherwise. In Fig.3 B, CD68 fluorescence is definitely decreased in the AZD-3965 treated group compared to control.

---

## Round 0.2 · accepted · Accept

· Academic Editor

Accept

Thanks for the thorough improvements to the manuscript, which is now accepted in PeerJ; Congratulations!

·

Basic reporting

The authors made sufficient changes to improve the overall quality of the manuscript.

Experimental design

The authors addressed the issue raised by this reviewer and made appropriate changes to experimental design.

Validity of the findings

The authors sufficiently addressed the issues raised and made improvements.

Additional comments

The authors sufficiently addressed all of the comments.